# WorkBench: a Benchmark Dataset for Agents in a Realistic Workplace Setting

**Olly Styles** *, **Sam Miller** *, **Patricio Cerda-Mardini**
Mindsdb
ollystyles@gmail.com, sam@artanis.ai, patricio@mindsdb.com

**Tanaya Guha**
University of Glasgow
tanaya.guha@glasgow.ac.uk

**Victor Sanchez**
University of Warwick
V.F.Sanchez-Silva@warwick.ac.uk

**Bertie Vidgen**
Contextual AI
bvidgen@turing.ac.uk

## Abstract

We introduce *WorkBench*: a benchmark dataset for evaluating agents' ability to execute tasks in a workplace setting. WorkBench contains a sandbox environment with five databases, 26 tools, and 690 tasks. These tasks represent common business activities, such as sending emails and scheduling meetings. The tasks in WorkBench are challenging as they require planning, tool selection, and often multiple actions. If a task has been successfully executed, one (or more) of the database values may change. The correct outcome for each task is unique and unambiguous, which allows for robust, automated evaluation. We call this key contribution *outcome-centric evaluation*. We evaluate five existing ReAct agents on WorkBench, finding they successfully complete as few as 3% of tasks (Llama2-70B), and just 43% for the best-performing (GPT-4). We further find that agents' errors can result in the wrong action being taken, such as an email being sent to the wrong person. WorkBench reveals weaknesses in agents' ability to undertake common business activities, raising questions about their use in high-stakes workplace settings. WorkBench is publicly available as a free resource at https://github.com/olly-styles/WorkBench.

## 1 Introduction

Large language models (LLMs) excel at a broad range of tasks such as translation, summarisation and sentiment analysis. However, they often fail at tasks such as retrieving niche knowledge or recent information (Bubeck et al., 2023). For example: they may give a wrong answer for this year's growth of the S&P 500. Proposed approaches to overcoming these limitations include finetuning and Retrieval Augmented Generation (Lewis et al., 2020), or RAG, which gives LLMs access to data stored as embedded vectors. However, these approaches cannot deal with other LLM failures, such as simple arithmetic problems (Brown et al., 2020). LLMs' inability to take actions also limits their capabilities. While they can summarise an email for a user, they cannot reply to it.

---

*Equal contributors

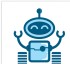

**User** 1:45pm
*Can you email Sarah to arrange a catch-up if we haven't met in the last week?*

**Agent** 1:46pm
*Sure! It's been 8 days since your last meeting with Sarah so I've sent her an email.*

Figure 1: **Agents in the workplace.** A sample task from WorkBench, the first dataset for evaluating autonomous agents on realistic workplace tasks.

Autonomous agents, powered by LLMs with access to tools, can overcome these limitations. Tools are functions that agents use to interact with their environment. Rather than relying solely on knowledge from training, agents can use tools such as calculators and search engines to broaden their capabilities. With access to external systems, these agents can perform a range of tasks such as replying to emails and booking meetings.

While agents are promising, they make mistakes and little is known about their efficacy in practice. Recent benchmark datasets cover tasks like solving logic problems (Mialon et al., 2023), or playing video games (Wang et al., 2023). These help understand agents' broader capabilities, but do not represent tasks an agent could be asked to perform in a workplace setting.

This paper introduces *WorkBench*: a benchmark dataset for evaluating agents in a realistic workplace setting. Tasks in WorkBench represent common functions across businesses, such as sending emails. WorkBench includes complex, multi-step tasks that require using several tools. For example: the task in Figure 1 requires an agent to review the user's calendar before sending an email.

For tasks in WorkBench, we introduce *outcome-centric evaluation*. Each task has a unique, unambiguous ground-truth outcome. Agents are evaluated on whether the outcome from their actions matches the ground truth outcome. This is a major step forward in enabling robust and automatic evaluation of action-taking agents.

We build a dataset of 690 unique tasks by combining human-curated task templates with a programmatic approach for creating multiple tasks per template. WorkBench tasks are challenging. We evaluate five different agents, finding that a state-of-the-art agent (ReAct with GPT-4) only completes 43% of tasks correctly.

To summarise, the contributions of this work are:

- An opensource dataset, WorkBench, which enables robust and automatic evaluation of agents in a realistic workplace setting.
- Our outcome-centric evaluation methodology for benchmarking agents.
- Benchmark results showing what current state-of-the-art agents are capable of.

## 2 Literature Review

**Tool Use Methods.** Many prior works have investigated using LLMs as a planning agent for API-based tools (Qin et al. (2023a) provide a survey). Augmenting LLMs with tools enables them to interact with external systems, so end-users can interact with systems through a chat interface. Closed-source LLMs, like GPT-4 (OpenAI et al., 2023), are assumed to be fine-tuned on tool-usage data, whereas opensource LLMs are generally not. This has led researchers to using closed-source LLMs to generate the training data for opensource LLMs (Liu et al., 2023).

Yao et al. (2023) propose the ReAct framework, which has been highly influential for agent design. This inspires approaches that i) transform the user's intent into a high-level task, ii) make a plan to achieve the task based on available tools, and finally iii) execute that plan (Qin et al., 2023b; Schick et al., 2023; Bubeck et al., 2023). Lu et al. (2023) show this approach is

effective when using tools across multiple domains, such as maths and information retrieval, with GPT-4 as the LLM.

Many papers build on the insight that adding examples of tool use to the prompt may improve performance (Schick et al., 2023). Hsieh et al. (2023) instead give tool documentation in the prompt, showing that this may be more effective than demonstrations. Hao et al. (2023) overcome this by embedding each tool in a separate space, then using this embedding space to choose tools. Song et al. (2023) show that first forming a natural language plan and then forming an explicit API call plan improves performance.

**Tool Use Limitations.** A limitation of some studies is assessing tool usage with GPT-3.5 as the only LLM, so others have analysed the impact of choosing different LLMs (Qin et al., 2023b; Hsieh et al., 2023; Schick et al., 2023). Ruan et al. (2023) find that most smaller LLMs cannot effectively use tools, particularly for multi-step problems, with only GPT-3 effectively using tools. Huang et al. (2023) support this conclusion, finding that most LLMs other than GPT-4 cannot effectively select the right tools on their benchmark dataset. This may be explained by the finding of Xu et al. (2023) that GPT-4 can make correct API calls without in-context examples, whereas most models cannot. This suggests tool usage is part of the training data for GPT-4. Nevertheless, Hao et al. (2023) propose improving opensource models by giving them access to embeddings of tool descriptions. Ruan et al. (2024) study the risk of negative side effects when agents are used in real-world setting, which we build on in this work.

**Tool Use Evaluation.** There are many datasets for evaluating LLM capabilities, such as Big-Bench (Srivavastava et. al, 2023), MMLU (Hendrycks et al., 2021) and HellaSwag (Zellers et al., 2019). These feature a range of topics such as coding, chess, and chemistry. However, they are of limited value for evaluating tool usage, given that questions are answerable without access to tools.

Other benchmarks have been proposed for specifically evaluating tool usage. Mialon et al. (2023) propose GAIA, which contains 466 questions that test agents on tasks such as web search and coding. Each question has a unique answer for robust, automatic evaluation. Patil et al. (2023) present APIBench for evaluating the effectiveness of LLM agents in calling other AI models accessed through APIs. SLURP (Bastianelli et al., 2020) and TaskMaster (Byrne et al., 2019) focus on smart home tasks that only require a single action. Other benchmarks use simulated environments, such as playing video games (Wang et al., 2023), human behaviour in a home (Park et al., 2023), and chat dialogues (Budzianowski et al., 2018).

Xu et al. (2023) propose ToolBench for evaluating the efficacy of LLMs in constructing real API calls for tasks such as travel booking. They generate a large volume of tasks with ChatGPT. Zhuang et al. (2023) propose ToolQA, which combines human task creation with a programmatic approach to scaling the dataset. However, both datasets are limited to information retrieval tasks. Li et al. (2023) propose API-Bank for evaluating agents using tools to take actions. Their automated approach enables a very large number of tools. However, answers are not unique or unambiguous. They require an LLM to evaluate performance of agents, therefore the validity of their evaluation depends on the evaluator LLM not making mistakes. Our proposed outcome-centric evaluation methodology overcomes this issue. Table 1 compares WorkBench against existing datasets.

**Web Browsing Agents.** Another relevant strand of literature is agents for web browsing. Their tools are implicitly defined via possible interactions with a browser (Deng et al., 2023; He et al., 2024; Yao et al., 2022). Some web browsing benchmarks use text-based evaluation, which shares properties with our evaluation methodology(Zhou et al., 2024; Koh et al., 2024). Particularly relevant and concurrent with our work, Drouin et al. (2024) also propose a robust method for evaluating retrieval tasks. Our benchmark builds on these papers by i) evaluating side effects from unintended state changes outside the target, and ii) focusing on workplace-relevant tasks.

| | Dataset | Tool usage | Actions | Outcome-centric evaluation |
|---|---|---|---|---|
| Datasets without tools | BigBench (Srivavastava et. al, 2023) | ✗ | ✗ | n/a |
| | MMLU (Hendrycks et al., 2021) | ✗ | ✗ | n/a |
| | HellaSwag (Zellers et al., 2019) | ✗ | ✗ | n/a |
| Retrieval-based datasets | GAIA (Mialon et al., 2023) | ✓ | ✗ | n/a |
| | ToolQA (Zhuang et al., 2023) | ✓ | ✗ | n/a |
| | ToolBench (Qin et al., 2023b) | ✓ | ✗ | n/a |
| Action-based datasets | API-Bank (Li et al., 2023) | ✓ | ✓ | ✗ |
| | WebArena (Zhou et al., 2024) | ✓ | ✓ | ✗ |
| | WorkArena (Drouin et al., 2024) | ✓ | ✓ | ✗ |
| | WorkBench (Ours) | ✓ | ✓ | ✓ |

Table 1: **Dataset comparison.** WorkBench is the first dataset built with outcome-centric evaluation. This enables robust, automatic evaluation of agents on tasks requiring actions.

## 3 Proposed Benchmark Dataset: WorkBench

Figure 2 shows our complete methodology for evaluating agents. Real-world agents typically interact with data using external API calls. However, we do not use data from real APIs such as Gmail, as these require authentication and change over time. Instead, we create a local environment with five sandbox databases. These represent the initial state of the agent's environment. In response to a task, the agent's actions can alter the sandbox databases. All tasks in WorkBench have a unique, unambiguous outcome, which is the expected state of the sandbox databases after successful completion of the task.

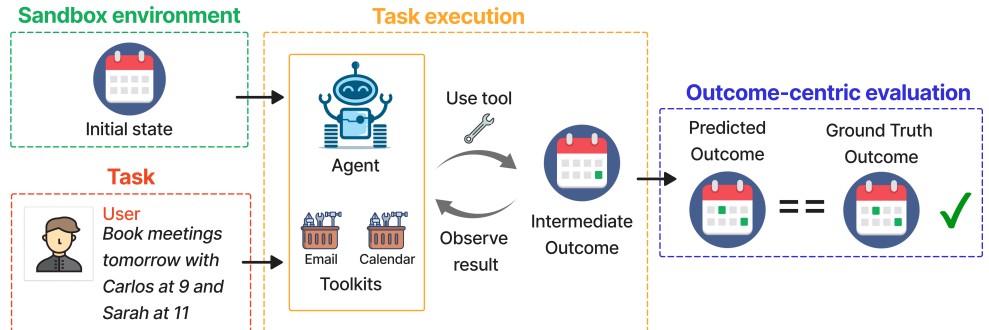

Figure 2: **Our complete pipeline for evaluating agents.** *1) Sandbox environment:* The sandbox has an initial state, defined by five databases. *2) Task*: a request is sent by the user. *3) Task execution:* a task is sent to the agent, which has access to toolkits in various domains. The agent takes actions using these tools, which may alter the sandbox databases. The agent observes the result of using the tool to determine if more actions are required. *4) Outcome-centric evaluation:* the updated sandbox databases are compared against the ground truth.

### 3.1 Sandbox Environment

We create a sandbox environment consisting of five simulated databases:

**Calendar.** Each event has an ID, name, participant email, start time and duration. There are 300 events in the database. See Table 2 for a snippet of Calendar data.

Figure 3: **Task and outcome creation.** The left side shows a pair of task-and-outcome templates. The outcome template is a function that returns the ground truth for the changes to the sandbox databases, given correct task completion. The right side shows a task-outcome pair created from these templates. In this example, the correct outcome is that the next meeting with Carlos is no longer in the Calendar sandbox.

**Email.** Each email has an ID, sender, subject, body, and datetime. There are 500 emails in the database.

**Website analytics.** Each website visit has a visitor ID, page view count, visit duration, traffic source, and visitor engagement score. There are 500 website visits in the database.

**Customer relationship management (CRM).** Each customer has an ID, name, email, phone number, activity status, assigned employee, date of last contact, product interest, follow up deadline, and notes. There are 200 customers in our database.

**Project management.** Each task has an ID, name, due date, assigned employee, list name, and board name. There are 300 tasks in our database.

| Event ID | Event Name | Participant Email | Start Time | Duration Minutes |
|---|---|---|---|---|
| 000013 | sync up | luis.ortiz@atlas.com | 2023-08-01 09:00 | 90 |
| 000275 | process review | fatima.khan@atlas.com | 2023-08-01 11:30 | 30 |
| 000264 | Onboarding | akira.sato@atlas.com | 2023-08-01 14:30 | 60 |

Table 2: **Calendar sandbox database snippet.** This is a sample of data from the Calendar sandbox database, which has a row for each event. We create sandbox databases for four other domains, which are further detailed in Appendix A.1

### 3.2 Task and outcome pairs

We manually create task templates that represent realistic workplace tasks. Templates are split into two categories: 1) Single domain: these only require tools from one domain, such as Calendar, to complete. 2) Multi-domain: these require tools from multiple domains, such as Email and Calendar, to complete.

Figure 3 shows how we programmatically create multiple tasks per template, and calculate the associated outcome. Across all domains, we create 10 unique tasks for each template, yielding 690 tasks in total. Table 3 shows the number of templates for each domain in WorkBench. Our templates contain added linguistic variation, so that each task is phrased in three different ways. We provide examples of this variation and empirical justification for our template approach in Appendix A.3.

We create some very challenging task templates by combining actions across multiple domains. Figure 4 shows the distribution of the number of actions required to complete

| Domain | Number of unique templates | Number of unique tasks |
|---|---|---|
| Analytics | 12 | 120 |
| Calendar | 11 | 110 |
| CRM | 8 | 80 |
| Email | 9 | 90 |
| Project Management | 8 | 80 |
| Multi-domain | 21 | 210 |
| **Total** | 69 | 690 |

Table 3: **Number of unique tasks and templates in each domain.** Our template approach creates 690 varied, realistic tasks. Each has an associated unique outcome.

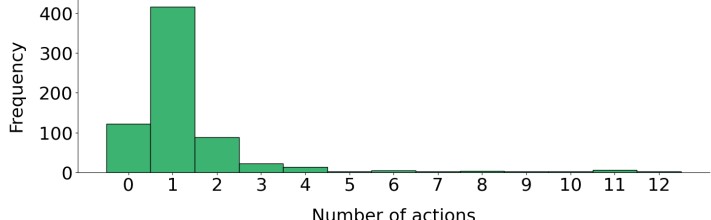

Figure 4: **Ground truth number of actions required to complete a task.** 18% of tasks require no actions. Sometimes agent would use retrieval tools, but would not need to execute any actions. For example: a request to cancel meetings on a date when there aren't any scheduled. These tasks are easier, which we show in Appendix A.4.

each task in our database. Many tasks require multiple actions, with some requiring up to 12 actions. Here is an example of a complex multi-domain task:

```
If our website page views fell by more than 10% in the past week, schedule a
30-minute meeting with Sam called "Urgent Analytics Update" at the earliest free
time tomorrow. Otherwise email them saying "Site traffic was stable the past week,
nice work."
```

We can define a simple function to automatically find the correct outcome. However, the agent must combine Analytics, Calendar and Email tools to complete this task. Appendix A.2 provides more examples of tasks.

### 3.3 Task Execution

Agents execute tasks using 26 tools across the five domains, which are summarised in Table 4. Each tool consists of a function that interacts with the sandbox databases, and documentation (a docstring) showing the agent how to use the tool. Each docstring contains a high-level description, parameters, return values, an example of tool usage, and any limits. An example limit is the search_events(start_time, end_time), which returns a maximum of five events. If the agent needs to find a large number of events, it must search multiple times with different time windows and then concatenate the results. Appendix A.5 contains the full docstring for each tool.

### 3.4 Outcome-Centric Evaluation

Figure 3 shows that the correct outcome, which is the ground truth, is always known. We then evaluate whether the outcome resulting from the agent's actions matches the ground truth. The ground truth therefore includes the state of all five sandbox databases. We call this methodology outcome-centric evaluation - Figure 5 compares our outcome-centric evaluation against prior works, which evaluate the agent's function calls.

| Email | Calendar | Web Analytics | CRM | Projects |
|-------|----------|---------------|-----|----------|
| `get_email_info` | `get_event_info` | `get_visitor_info` | `get_customer_info` | `get_task_info` |
| `search_emails` | `search_events` | `count_traffic_source` | `search_customers` | `search_tasks` |
| `send_email` | `create_event` | `count_engaged_users` | `update_customer` | `create_task` |
| `delete_email` | `delete_event` | `count_total_visits` | `add_customer` | `delete_task` |
| `forward_email` | `update_event` | `average_visit_duration` | `delete_customer` | `update_task` |
| `reply_email` | | `create_plot` | | |

Table 4: **Summary of toolkits.** We define 26 tools across 5 domains.

An agent can follow any action path provided the resulting sandbox databases match the ground truth outcome. For example, sometimes the agent recovers from its error and takes the correct action:

**Task:**
*Make a task on the Front end board for Sam to improve conversion.*
**Ground truth tool use:**
`create_task(name="improve conversion", board="Front end", assigned_to="Sam")`
**Agent's tool use:**
`create_task(name="improve conversion", board="Front End", assigned_to="Sam")`
`Observation: "'Front End' board does not exist, but 'Front end' does..."`
`create_task(name="improve conversion", board="Front end", assigned_to="Sam")`

Evaluation methods based on matching the function calls could unfairly find the agent had failed due to the extra calls. However, outcome-centric evaluation recognises that the agent was able to recover because the final change in state matches the ground truth outcome. As a result, the agent is not unfairly penalised.

Some prior benchmarks such as Gaia (Mialon et al., 2023) and ToolQA (Zhuang et al., 2023) also have tasks with unique outcomes, but they are limited to information retrieval. WorkBench is the first dataset to evaluate tasks that require actions in this manner, due to our outcome-centric evaluation methodology.

# 4 Results

We assess the performance of LLM agents using the ReAct framework (Yao et al., 2023). This enables the LLM to perform multiple action steps and update its action plan based on results from previous steps.

## 4.1 Performance Metrics

Our primary metric is *accuracy*. This is the % of tasks where the outcome from the agent's actions match the expected outcome, which is the ground truth.

Our secondary metric is *side effects*[1]. Some tools have negative consequences if used incorrectly, such as sending emails to the wrong person. If the agent's actions modify the sandbox databases in a way that does not match the ground truth exactly, we consider this a side effect. If the agent fails to complete the task, but does not alter the sandbox databases, then there are no side effects.

## 4.2 Comparing Large Language Models

Table 5 compares five LLMs: GPT-3.5 (Brown et al., 2020), GPT-4 (OpenAI et al., 2023), Claude-2 (Anthropic, 2023), Llama2-70B (Touvron et al., 2023) and Mixtral-8x7B (Jiang et al., 2024). GPT-4 greatly outperforms other models. For the worse-performing models, the main errors are insufficient context window length and failing to follow the ReAct framework.

---

[1]This computer science term refers to a program altering variables outside its local environment.

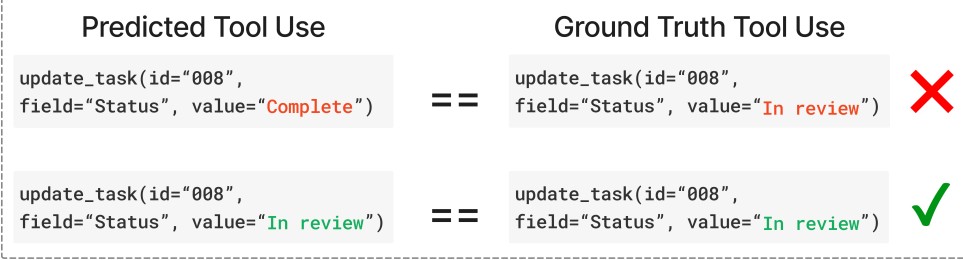

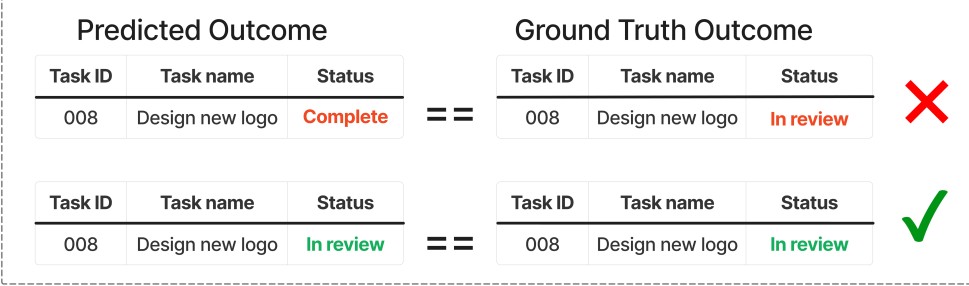

Figure 5: **Outcome-Centric Evaluation.** We propose outcome-centric evaluation, where there is a unique ground-truth outcome for each task (lower panel). We consider the task correctly executed if the predicted outcome following the agent's actions matches this outcome. This allows the agent to find multiple paths to the correct outcome, unlike prior works (upper panel) which evaluate the agent's function calls.

|                           | GPT-4 | GPT-3.5 | Claude-2 | Llama2-70B | Mixtral-8x7B |
|---------------------------|-------|---------|----------|------------|--------------|
| Accuracy (required tools) | 49%   | 14%     | 23%      | 3%         | 20%          |
| Accuracy (all tools)      | 43%   | 0%      | 26%      | 0%         | 16%          |

Table 5: **Comparing accuracy across all models on WorkBench.** We selected these models based on general usage and prevalence in other benchmark studies. Due to limits on context window size, the upper row shows when models are given only the toolkit from the domain needed to complete each task. For example: if the task were "Book a 30-minute meeting with Sam for tomorrow at 9:30", we would provide all tools in the calendar toolkit and none from other toolkits. The lower row is when models are given all 26 tools. Scores of 0% occur when the context window is not long enough to include all 26 tool descriptions.

Our benchmark is challenging for all models, including GPT-4. Given how poorly other models perform, we restrict further analysis to GPT-4. When giving this agent all 26 tools, rather than just the required toolkits, and find accuracy falls from 49% to 43%. This suggests the agent is negatively affected by redundant tools, which is consistent with prior studies (Hao et al., 2023). The next sections explore in further depth why the GPT-4 agent fails.

### 4.3 Performance across domains

Table 6 compares the GPT-4 agent's performance across our five individual domains, and tasks that require tools from multiple domains. Performance varies from 23% accuracy on CRM tasks to 65% for Calendar tasks. The agent is capable of combining tools across multiple domains. Its performance (40%) on these tasks is similar to its average performance on single-domain tasks (43%).

|  | Analytics | Calendar | CRM | Email | Project Management | Multi Domain |
|---|---|---|---|---|---|---|
| Number of tasks | 120 | 110 | 80 | 90 | 80 | 210 |
| Accuracy (↑) | 39% | 65% | 23% | 48% | 39% | 40% |
| Side Effects (↓) | 54% | 22% | 6% | 6% | 4% | 29% |

Table 6: **Performance of GPT-4 ReAct agent on WorkBench with all tools provided.**. Higher is better (↑) for accuracy and lower is better (↓) for side effects. Side effects are particularly common in the Analytics domain, as the agent often plots data for the wrong time period.

## 4.4 Sources of Error

Figure 6a shows the prevalence of side effects. Side effects occur when the agent's actions modify the sandbox environment, but this change does not match the ground truth outcome. In this example from a task in our dataset, the agent cancels the wrong meeting:

**Task:** *Cancel my next meeting with Nadia*
**Ground truth:** `delete_event(event_id=00000035)`
**Prediction:** `delete_event(event_id=00000196)`

Errors without side effects occur when the agent fails to complete a task, but there are no unintended modifications to the environment. In this example, the agent does not take any actions because it searches on the wrong date:

*(Today's date, Monday 20th November)*
**Task**: Cancel all my meetings on Tuesday
**Ground Truth**: `delete_event(event_id=00000025)`
**Prediction**: Searches for meetings on 28th November - no meetings are found

Figures 6b and 6c further break down errors by their most common sources. The most frequent error is failing to follow the ReAct framework. The agent must use the keyword `ACTION` followed by a `JSON` string with the tool name and arguments. The agent may omit the `ACTION` keyword, meaning no actions are performed.

To address this source of error, we implement a re-sampling strategy. If the agent does not correctly generate the `ACTION` keyword, we repeat the same task up to 5 times. We use a temperature of 0.5 for retries, rather than the initial temperature of 0. With this strategy, the accuracy of the GPT-4 agent increases from 43% to 49%.

However, other common sources of error cannot be fixed with resampling. The agent often fails to find the correct email address when given names in the task. The agent may hallucinate an email address rather than using the search tool to find the correct email address. Here is an example, with intermediate steps hidden for concision:

**Task:** *Forward all the emails from kofi last week about 'Staff Roster for Next Week' to fatima*
**Ground truth:** `forward_email(email_id="0249",recipient="fatima.khan@atlas.com")`
**Prediction:** `forward_email(email_id="0249", recipient="fatima@example.com")`

Errors also come from searching incorrectly. The agent may fail on tasks such as *"Cancel my next meeting with Sam"* because it searches for events in the past. Similarly, the agent does not always account for the limits of its tools. It can fail on tasks such as *"Cancel all future meetings with Sam"* because it only cancels a subset of future events after failing to factor in the limit on the number of results returned from searches. The agent could complete this task by repeating the search-and-deletion process, but fails to do so.

## 5 Discussion and Future Work

We have introduced WorkBench - the first benchmark that enables robust, automatic evaluation of agents in a workplace setting. Our approach ensures that each task has a unique

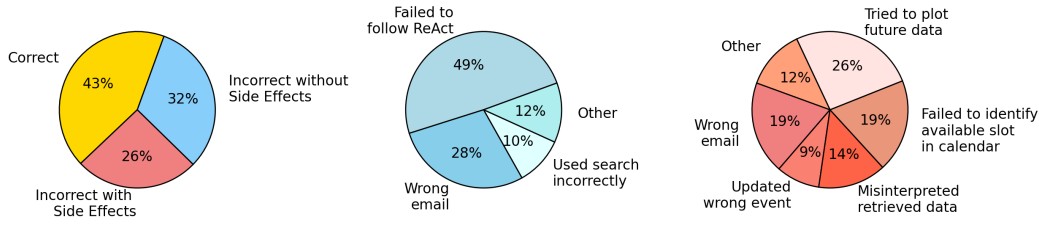

(a) Overall Error Breakdown    (b) Errors without Side Effects    (c) Errors with Side Effects

Figure 6: **Error breakdown for GPT-4 across all domains with all tools provided.** Breakdown of errors with side effects vs those with no side effects (left). Detailed breakdown of errors with no side effects (middle) and errors with side effects (right).

outcome, which is the expected change to the state of the sandbox environment upon successful task completion.

One limitation of WorkBench is how well the sandbox environment represents real-world complexity. A real email inbox may contain tens of thousands of emails over many years, and emails are often spam, very long, and/or full of errors. Our initial results may therefore overestimate agents' current capabilities, so further studies could improve WorkBench by adding more challenging sandbox data.

While our tasks require multiple actions, they are limited to single-turn chat. Some longer queries may not represent realistic workplace interactions. Establishing a human baseline on our tasks would help determine the significance of this, but is hard to implement given our tasks would be completed by humans using a Graphical User Interface (GUI). Implementing a human baseline would measure GUI quality as well as task difficulty. Nevertheless, a multi-turn chat setup may be more representative of real tasks and could build upon our work.

We also found the agent's performance dropped when it had a greater number of tools it could choose from, but we were not able to explore this relationship fully with just 26 tools. Future work could extend our benchmark by adding more tools from other real-world domains such as HR software. This would help assess the relationship between agent accuracy and breadth of tooling.

A final limitation is that we do not assess pure retrieval tasks. While retrieval tools are a required intermediate step to complete many WorkBench tasks, we do not assess tasks that require solely retrieval such as finding a recent email. Future papers could build on our work with a method for assessing retrieval tasks, such as those proposed by other recent benchmarks (Drouin et al., 2024; Zhuang et al., 2023).

Despite these limitations, WorkBench has a large volume of high-quality, unique tasks. We include complex tasks that require planning, tool selection and analysing results across five domains. WorkBench is challenging, with the best agent achieving only 43% accuracy. We find the main sources of error are the agent failing to execute its plan, giving the wrong arguments to tools, and not understanding the limits of its tools. Furthermore, errors often have negative consequences like sending emails to the wrong people. Future work could study fine-tuning LLMs to improve performance, such as (Schick et al., 2023).

WorkBench is both scalable and extensible. Future researchers could extend our dataset to include new domains, such as accounting tasks, and build an even larger dataset using our scalable approach to task creation. This will enable the evaluation of agents in progressively more complex settings as they continue to improve in the future.

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

## A  Appendix

### A.1  Simulated database creation

Below is a sample of 5 rows from each of our 5 sandbox databases.

### A.1.1  Analytics

```
date_of_visit,visitor_id,page_views,session_duration_seconds,traffic_source,user_engaged
2023-10-22,860,8,4,referral,False
2023-11-22, 214, 11, 1, search engine, False
2023-09-24, 130, 18, 0, social media, False
2023-10-08, 385, 2, 4, direct, False
2023-09-22, 252, 2, 11, search engine, False
```

### A.1.2  Calendar

```
event_id, event_name, participant_email, event_start, duration
00000013, sync up, luis.ortiz@atlas.com, 2023-08-01 09:00:00, 90
00000275, process review, fatima.khan@atlas.com, 2023-08-01 11:30:00, 90
00000098, Data Security and Compliance Training, amir.ali@atlas.com, 2023-08-02
11:00:00, 30
00000190, Product Launch Analysis, yuki.tanaka@atlas.com, 2023-08-02 11:30:00,
30
00000071, daily stand-up, kofi.mensah@atlas.com, 2023-08-02 13:30:00, 30
```

### A.1.3  Customer Relationship Manager

**customer_id, assigned_to_email, customer_name, customer_email, customer_phone, last_contact_date, product_interest, status, follow_up_by, notes**
```
00000189, lena.schmidt@atlas.com, Taylor Jackson, taylor.jackson@nanolabs, ,
2023-11-30, Consulting, Lost, 2023-12-22, 2023-11-07: Had a call. 2023-11-25:
Had a call.
00000107, sofia.santos@atlas.com, Quinn Harris, quinn.harris@nanoforcerobotics,
, 2023-11-30, Consulting, Proposal, 2023-12-14, 2023-11-26: Saw the demo.
2023-11-29: Had a call. 2023-10-27: Had a call.
00000052, raj.patel@atlas.com, Jaden White, jaden.white@protracefoods,
724-857-2625, 2023-11-30, Hardware, Won, 2023-12-13, 2023-10-17: Had a call.
00000102, sofia.santos@atlas.com, Alex Thomas, alex.thomas@proenergy, ,
2023-11-30, Hardware, Qualified, 2023-12-22, 2023-10-15: On holiday.
00000187, lena.schmidt@atlas.com, Quinn Robinson, quinn.robinson@flexenergy,
399-396-5380, 2023-11-30, Hardware, Lead, 2023-12-23,
```

### A.1.4  Email

**email_id, inbox/outbox, sender/recipient, subject, sent_datetime, body**
```
00000373, inbox, santiago.martinez@atlas.com, Task Update on Develop prototype for
payment gateway, 2023-10-01 09:15:02, "Sam, \n Completed task 'Develop prototype
for payment gateway' ahead of schedule. Please review and let me know if any
tweaks are needed.\n\n Best,\n Santiago"
00000353, inbox, chenwei.zhang@atlas.com, Update on Annual Budget Planning
Session, 2023-10-01 09:40:01, "Sam,\n Encountered a few challenges while working
on the Annual Budget Planning Session. Could use your advice.\n\n Cheers,\n
Chenwei" 00000013, inbox, kofi.mensah@atlas.com, Task Update on Fix alignment
issue in homepage, 2023-10-01 10:50:46, "Dear Sam, \n Regarding task 'Fix
alignment issue in homepage', I've made significant progress but have hit a
snag with third-party API compatibility. Could use a brainstorm session.\n \n
Regards, \n Kofi"
00000103, inbox, chenwei.zhang@atlas.com, Update on Quarterly Sales Review,
2023-10-01 10:58:07, "Hey Sam, \n Encountered a few challenges while working
on the Quarterly Sales Review. Could use your advice.\n \n Thanks, \n Chenwei"
00000295, inbox, nadia.moreau@atlas.com, Update on Year-End Performance
Assessment, 2023-10-01 11:37:37, "Hey Sam, \n Could you provide your input on the
Year-End Performance Assessment planing? Your insights would be really valuable.
\n \n Additionally, I wanted to touch base on some other areas we've been focusing
on lately. Our team has been working tirelessly on improving our project management
workflows and enhancing collaboration across departments. This effort includes
adopting new tools, refining our communication strategies, and ensuring that all
team members are fully aligned with our objectives. \n \n Best, \n Nadia"
```

### A.1.5 Project management

**task_id, task_name, assigned_to_email, list_name, due_date, board**
00000149, Add animation to carousel, leila.azizi@atlas.com, Backlog, 2023-11-28, Front end
00000037, Add authentication for email notification, carlos.rodriguez@atlas.com, Backlog, 2023-11-28, Back end
00000061, Update Flask to latest version, aisha.chen@atlas.com, Backlog, 2023-11-28, Back end
00000093, Optimize database query for search functionality, fatima.khan@atlas.com, Backlog, 2023-11-28, Back end
00000096, Add authentication for third-party login, carlos.rodriguez@atlas.com, Backlog, 2023-11-28, Back end

## A.2 Example tasks

The following are 5 randomly sampled tasks from each domain and 5 randomly sampled multi-domain tasks.

### A.2.1 Analytics

- Please plot for me the distribution of engaged users and average session duration between October 14 and November 6
- Can you make a line plot of the most popular traffic source since November 27?
- Was total visits more than 10 at any time in the last 2 weeks? If so, please plot it as a line chart
- Can you plot the distribution of both total visits and average session duration between October 12 and November 6?
- Can you make a line plot of the most popular traffic source since October 15?

### A.2.2 Calendar

- Create a 1.5 hour event called New Employee Onboarding on December 8 at 3:30 with nia
- Cancel my next meeting with yuki
- Delete the next Annual Budget Planning Session meeting
- have I met with carlos in the last 7 days? If not, schedule a 30-minute meeting called 'catch-up' for my first free slot from tomorrow
- something came up. Can you cancel my meetings on Friday before 10:30?

### A.2.3 Customer Relationship Manager

- Give Sofia all of Lena's customers that are interested in training and are either qualified or in proposal in the crm
- I need to move all of Sofia's customers that are interested in training and are either qualified or in proposal to Nadia. Can you make that change in the crm?
- Reassign all of Nadia's leads that are interested in training to Lena in the crm.
- Move all customers that haven't responded to a proposal for the consulting product in 5 weeks to lost in the crm
- Sofia is taking over all of Lena's customers that are interested in services and are either qualified or in proposal. Can you reassign them in the crm?

### A.2.4  Email

- I need to reply to the latest email from kofi with 'Got it, thank you!'. Can you do that?
- can you forward the latest email about 'Task Update on Design logo for blog' to carlos
- lena and aisha need the last email about 'Update on Team Building Retreat'. Can you forward it?
- Reply to yuki's last email about 'Update on Corporate Social Responsibility Initiative' with 'Thanks for the update - I will get back to you tomorrow.
- Delete my last email from chenwei

### A.2.5  Project Management

- Move any of luis's tasks that are in review to completed
- Give all the overdue tasks that fatima hasn't started to santiago
- Move any of nia's tasks that are in review to completed
- Give all the overdue tasks that chenwei hasn't started to amir.
- can you move any of luis's tasks that are in review to completed?

### A.2.6  Multi-domain

- I need to make sure everyone remembers to attend the first event on December 6. Can you send an email to the attendees with the event name as the title and 'Remember to attend this event.' in the email?
- I need to make sure everyone remembers to attend the first event on December 1. Can you send an email to the attendees with the event name as the title and 'Remember to attend this event.' in the email?
- please check the percent growth of engaged users since Friday. If it grew by more than average session duration make a front-end backlog task called 'Improve average session duration' for kofi that's due next Friday and schedule a 30 minute meeting called 'Discuss engaged users' for us at the earliest slot i'm free tomorrow
- I think carlos might have some overdue tasks. Can you check and if so, send them an email titled 'Overdue tasks' saying 'You have a few overdue tasks - can you update me on them?'. Otherwise email them with 'Nice work keeping on top of your tasks this sprint!' titled 'Good work this sprint'
- if fatima hasn't sent me any emails in the past 3 days, schedule a half hour meeting with them for Friday at 12 and call it 'Catch up with fatima'

### A.3  Template dependence

As introduced in Section 3.2, we use templates to create a large number of tasks. Figure 7 shows the percentage of tasks within each template that were completed correctly by the GPT-4 agent. This varies between 10% and 90% for most tasks, indicating that our approach creates a diverse set tasks for each template.

We introduce linguistic variation by writing three templates per tasks. The following is an example of the three templates used for a task in the calendar domain:

- Cancel all future meetings with {name}
- {name} is leaving the company. Can you cancel all future meetings with them?
- I need to cancel all future meetings with {name}. Can you do that for me please?

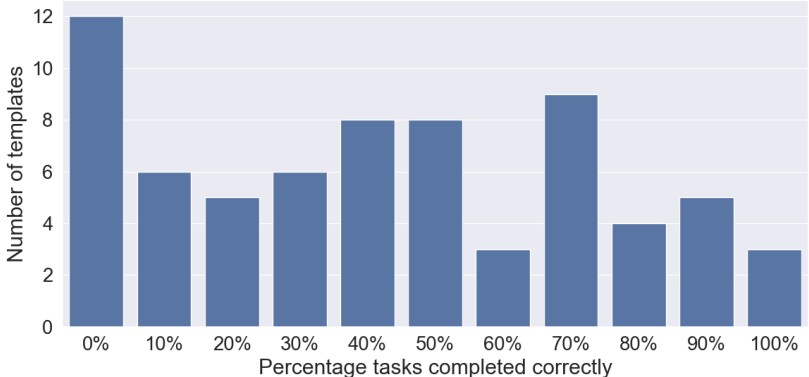

Figure 7: **Template dependence.** Our approach to task creation using templates creates a diverse set of tasks. The agent achieves 0% or 100% accuracy on just 15 out of 69 templates, indicating that tasks within a template are sufficiently diverse to evaluate agent performance.

## A.4 Impact of "no action" tasks

As shown in Figure 4, 122 of the 690 tasks in WorkBench do not require any actions to complete. The following is an example of such a task:

```
If I haven't met with akira in the last 7 days, schedule a 30-minute meeting
called 'catch-up' for my first free slot from tomorrow
```

Using the `calendar.search_events` tool, the agent can determine that the condition for scheduling a meeting is not met and therefore no action is required. In Table 7, we compare the accuracy using subsets of WorkBench based on the number of actions required. All tools are provided in the prompt.

|                           | GPT-4 | GPT-3.5 | Claude-2 | Llama2-70B | Mixtral-8x7B |
|---------------------------|-------|---------|----------|------------|--------------|
| Accuracy (all tasks)      | 43%   | 0%      | 24%      | 0%         | 16%          |
| Accuracy (0 action tasks) | 75%   | 0%      | 71%      | 0%         | 77%          |
| Accuracy (1+ action tasks)| 36%   | 0%      | 14%      | 0%         | 2%           |
| Accuracy (2+ action tasks)| 18%   | 0%      | 1%       | 0%         | 1%           |

Table 7: **Performance comparision based on number of actions.** All agents perform better on tasks that require no actions. Note that all tools are provided in the prompt, which cannot fit into the context window for GPT3.5 and Llama2-70B.

## A.5 Prompt and tool descriptions

The following is the full prompt template provided to the agent. The description of each tool provided to the agent is in the prompt.

```
Today's date is Thursday, 2023-11-30 and the current time is 00:00:00. Remember
the current date and time when answering queries. Meetings must not start
before 9am or end after 6pm.Respond to the human as helpfully and accurately as
possible. You have access to the following tools:

email.get_email_information_by_id: email.get_email_information_by_id(
email_id=None, field=None) - Retrieves specific details of an email by its ID.

Parameters    ----------    email_id : str, optional      Unique ID of the
email.    field : str, optional       Specific field to return. Available
```

fields: "email_id", "sender", "subject", "sent_date", "body", "inbox/outbox".

Returns      -------     email_information : dict        Information of the specified email for the given ID and field.

Examples     --------     >>> email.get_email_information_by_id("12345678", "subject")    {{"subject": "Project Update"}}

>>> email.get_email_information_by_id("12345678", "sent_date")     {{ "sent_date": "2024-01-10 09:30:00"}}, args: {{'email_id': {{'title': 'Email Id'}}, 'field': {{'title': 'Field'}}}}

email.search_emails: email.search_emails(query='', date_min=None, date_max=None) - Searches for emails matching the given query across subject, body, or sender fields.

The function matches an email if all words in the query appear in any of these fields.

Parameters

----------

query : str, optional

Search query, matching terms in subject, body, or sender fields.

date_min : str, optional

Lower date limit for the email's sent date (inclusive). Format: "YYYY-MM-DD"

date_max : str, optional

Upper date limit for the email's sent date (inclusive). Format: "YYYY-MM-DD"

Returns

-------

emails : list

List of emails matching the query criteria.

Examples

--------

>>> email.search_emails("Project Update")

[{{"email_id": "12345678", "inbox/outbox": "inbox", "subject": "Project Update", "sender/recipient": "jane@example.com", "sent_datetime": "2024-01-10 09:30:00", "body": "Please find the project update attached."}}], args: {{ 'query': {{'title': 'Query', 'default': ''}}, 'date_min': {{'title': 'Date Min'}}, 'date_max': {{'title': 'Date Max'}}}}

email.send_email: email.send_email(recipient=None, subject=None, body=None) -

Sends an email to the specified recipient.

Parameters

----------

recipient : str, optional

Email address of the recipient.

subject : str, optional

Subject line of the email.

body : str, optional

Body content of the email.

Returns

-------

message : str

Confirmation message of the email being sent.

Examples

--------

>>> email.send_email("jane@example.com", "Meeting Reminder", "Don't forget our meeting at 10am tomorrow.")

"Email sent successfully.", args: {{'recipient': {{'title': 'Recipient'}}, 'subject': {{'title': 'Subject'}}, 'body': {{'title': 'Body'}}}}

email.delete_email: email.delete_email(email_id=None) - Deletes an email by its ID.

Parameters

----------

email_id : str, optional

Unique ID of the email to be deleted.

Returns

-------

message : str

Message indicating whether the deletion was successful.

Examples

--------

```
>>> email.delete_email("12345678")
```

"Email deleted successfully.", args: {{'email_id': {{'title': 'Email Id'}}}}

email.forward_email: email.forward_email(email_id=None, recipient=None) -
Forwards an email to the specified recipient.

Parameters

----------

email_id : str, optional

Unique ID of the email to be forwarded.

recipient : str, optional

Email address of the recipient.

Returns

-------

message : str

Message indicating whether the email was forwarded successfully.

Examples

--------

```
>>> email.forward_email("12345678", "jane@example.com")
```

"Email forwarded successfully.", args: {{'email_id': {{'title': 'Email Id'}},
'recipient': {{'title': 'Recipient'}}}}

email.reply_email: email.reply_email(email_id=None, body=None) - Replies to an
email by its ID.

Parameters

----------

email_id : str, optional

Unique ID of the email to be replied.

body : str, optional

Body content of the email.

Returns

-------

message : str

Confirmation message of the email being replied.

Examples

--------

>>> email.reply_email("12345678", "Thank you for the update.")

"Email replied successfully.", args: {{'email_id': {{'title': 'Email Id'}}, 'body': {{'title': 'Body'}}}}

calendar.get_event_information_by_id: calendar.get_event_information_by_id( event_id=None, field=None) - Returns the event for a given ID.

Parameters

----------

event_id : str, optional

8-digit ID of the event.

field : str, optional

Field to return. Available fields are: "event_id", "event_name", "participant_email", "event_start", "duration"

Returns

-------

event : dict

Event information for the given ID and field.

Examples

--------

>>> calendar.get_event_information_by_id("00000000", "event_name")

{{"event_name": "Meeting with Sam"}}

>>> calendar.get_event_information_by_id("00000000", "event_start")

```
{{"event_start": "2021-06-01 13:00:00"}}
```

```
>>> calendar.get_event_information_by_id("00000000", "duration")
```

```
{{"duration": "60"}}, args: {{'event_id': {{'title': 'Event Id'}}, 'field': {{
'title': 'Field'}}}}
```

```
calendar.search_events: calendar.search_events(query='', time_min=None,
time_max=None) - Returns the events for a given query.
```

```
Parameters
```

```
----------
```

```
query: str, optional
```

```
Query to search for. Terms will be matched in the event_name and
participant_email fields.
```

```
time_min: str, optional
```

```
Lower bound (inclusive) for an event's end time to filter by. Format: "YYYY-MM-
DD HH:MM:SS"
```

```
time_max: str, optional
```

```
Upper bound (inclusive) for an event's start time to filter by. Format: "YYYY-
MM-DD HH:MM:SS
```

```
Returns
```

```
-------
```

```
events : list
```

```
List of events matching the query. Returns at most 5 events.
```

```
Examples
```

```
--------
```

```
>>> calendar.search_events("Sam")
```

```
[{{"event_id": "00000000", "event_name": "Meeting with Sam",
"participant_email: "sam@example.com", "event_start": "2021-06-01 13:00:00",
"duration": "60"}},
```

```
{{"event_id": "00000001", "event_name": "Lunch with Sam", "participant_email":
"sam@example.com", "event_start": "2021-06-01 13:00:00", "duration": "30}}"
```

```
], args: {{'query': {{'title': 'Query', 'default': ''}}, 'time_min': {{'title':
'Time Min'}}, 'time_max': {{'title': 'Time Max'}}}}
```

```
calendar.create_event: calendar.create_event(event_name=None,
```

participant_email=None, event_start=None, duration=None) - Creates a new event.

Parameters

----------

event_name: str, optional

Name of the event.

participant_email: str, optional

Email of the participant.

event_start: str, optional

Start time of the event. Format: "YYYY-MM-DD HH:MM:SS"

duration: str, optional

Duration of the event in minutes.

Returns

-------

event_id : str

ID of the newly created event.

Examples

--------

>>> calendar.create_event("Meeting with Sam", "sam@example.com", "2021-06-01 13:00:00", "60")

"00000000", args: {{'event_name': {{'title': 'Event Name'}}, 'participant_email': {{'title': 'Participant Email'}}, 'event_start': {{'title': 'Event Start'}}, 'duration': {{'title': 'Duration'}}}}

calendar.delete_event: calendar.delete_event(event_id=None) - Deletes an event.

Parameters

----------

event_id: str, optional

8-digit ID of the event.

Returns

-------

message : str

Message indicating whether the deletion was successful.

Examples

--------

>>> calendar.delete_event("00000000")

"Event deleted successfully.", args: {{'event_id': {{'title': 'Event Id'}}}}

calendar.update_event: calendar.update_event(event_id=None, field=None, new_value=None) - Updates an event.

Parameters

----------

event_id: str, optional

8-digit ID of the event.

field: str, optional

Field to update.

new_value: str, optional

New value for the field.

Returns

-------

message : str

Message indicating whether the update was successful.

Examples

--------

>>> calendar.update_event("00000000", "event_name", "New Event Name")

"Event updated successfully.", args: {{'event_id': {{'title': 'Event Id'}}, 'field': {{'title': 'Field'}}, 'new_value': {{'title': 'New Value'}}}}

analytics.engaged_users_count: analytics.engaged_users_count(time_min=None, time_max=None) - Returns the number of engaged users within a specified time range.

Parameters

----------

time_min : str, optional

Start date of the time range. Date format is "YYYY-MM-DD".

time_max : str, optional

End date of the time range. Date format is "YYYY-MM-DD".

Returns

-------

engaged_users : dict

Number of engaged users in the specified time range.

Examples

--------

>>> analytics.engaged_users_count("2023-10-01", "2023-10-06")

{{"2023-10-01": 1, "2023-10-02": 2, "2023-10-03": 2, "2023-10-04": 1, "2023-10-
05": 0, "2023-10-06": 4}}, args: {{'time_min': {{'title': 'Time Min'}},
'time_max': {{'title': 'Time Max'}}}}}

analytics.get_visitor_information_by_id:
analytics.get_visitor_information_by_id(visitor_id=None) - Returns the
analytics data for a given visitor ID.

Parameters

----------

visitor_id : str, optional

ID of the visitor.

Returns

-------

visitor_data : dict

Analytics data for the given visitor ID.

Examples

--------

>>> analytics.get_visitor_information_by_id("000")

{{"date_of_visit": "2023-10-01", "visitor_id": "000", "page_views": "3",
"session_duration_seconds": "10.0", "traffic_source": "search engine",
"user_engaged": "False"}}, args: {{'visitor_id': {{'title': 'Visitor Id'}}}}}}

analytics.traffic_source_count: analytics.traffic_source_count(time_min=None,
time_max=None, traffic_source=None) - Returns the number of visits from a
specific traffic source within a specified time range.

Parameters

----------

time_min : str, optional

Start date of the time range. Date format is "YYYY-MM-DD".

time_max : str, optional

End date of the time range. Date format is "YYYY-MM-DD".

traffic_source : str, optional

Traffic source to filter the visits. Available values are: "direct",
"referral", "search engine", "social media"

Returns

-------

traffic_source_visits : dict

Number of visits from the specified traffic source in the specified time range.

Examples

--------

>>> analytics.traffic_source_count("2023-10-01", "2023-10-06", "search engine")

{{"2023-10-01": 0, "2023-10-02": 1, "2023-10-03": 0, "2023-10-04": 3, "2023-10-
05": 2, "2023-10-06": 4}}, args: {{'time_min': {{'title': 'Time Min'}},
'time_max': {{'title': 'Time Max'}}, 'traffic_source': {{'title': 'Traffic
Source'}}}}}}

analytics.total_visits_count: analytics.total_visits_count(time_min=None,
time_max=None) - Returns the total number of visits within a specified time
range.

Parameters

----------

time_min : str, optional

Start date of the time range. Date format is "YYYY-MM-DD".

time_max : str, optional

End date of the time range. Date format is "YYYY-MM-DD".

Returns

-------

total_visits : dict

Total number of visits in the specified time range.

Examples

--------

>>> analytics.total_visits_count("2023-10-01", "2023-10-06")

{{"2023-10-01": 1, "2023-10-02": 2, "2023-10-03": 3, "2023-10-04": 1, "2023-10-05": 0, "2023-10-06": 4}}, args: {{'time_min': {{'title': 'Time Min'}}, 'time_max': {{'title': 'Time Max'}}}}

analytics.create_plot: analytics.create_plot(time_min=None, time_max=None, value_to_plot=None, plot_type=None) - Plots the analytics data for a given time range and value.

Parameters

----------

time_min : str, optional

Start date of the time range. Date format is "YYYY-MM-DD".

time_max : str, optional

End date of the time range. Date format is "YYYY-MM-DD".

value_to_plot : str, optional

Value to plot. Available values are: "total_visits", "session_duration_seconds", "user_engaged", "direct", "referral", "search engine", "social media"

plot_type : str, optional

Type of plot. Can be "bar", "line", "scatter" or "histogram"

Returns

-------

file_path : str

Path to the plot file. Filename is {{time_min}}_{{time_max}}_{{value_to_plot}}
_{{plot_type}}.png.

Examples

--------

>>> analytics.create_plot("2023-10-01", "2023-12-31", "total_visits")

"plots/2023-10-01_2023-12-31_total_visits.png", args: {{'time_min': {{'title':
'Time Min'}}, 'time_max': {{'title': 'Time Max'}}, 'value_to_plot': {{'title':
'Value To Plot'}}, 'plot_type': {{'title': 'Plot Type'}}}}

analytics.get_average_session_duration: analytics.get_average_session_duration(
time_min=None, time_max=None) - Returns the average session duration within a
specified time range.

Parameters

----------

time_min : str, optional

Start date of the time range. Date format is "YYYY-MM-DD".

time_max : str, optional

End date of the time range. Date format is "YYYY-MM-DD".

Returns

-------

average_session_duration : float

Average session duration in seconds in the specified time range.

Examples

--------

>>> analytics.get_average_session_duration("2023-10-01", "2023-10-06")

{{"2023-10-01": 10.0, "2023-10-02": 20.5, "2023-10-03": 32.8, "2023-10-04":
40.2, "2023-10-05": 5.3, "2023-10-06": 53.0}}, args: {{'time_min': {{'title':
'Time Min'}}, 'time_max': {{'title': 'Time Max'}}}}

project_management.get_task_information_by_id:
project_management.get_task_information_by_id(task_id=None, field=None) -
Returns the task infomration for a given ID.

Parameters

```
----------

task_id : str, optional

8-digit ID of the task.

field : str, optional

Field to return. Available fields are: "task_id", "task_name",
"assigned_to_email", "list_name", "due_date", "board"

Returns

-------

task : dict

Task information for the given ID and field.

Examples

--------

>>> project_management.get_task_information_by_id("00000000", "task_name")

{{"task_name": "Refactor code"}}, args: {{'task_id': {{'title': 'Task Id'}},
'field': {{'title': 'Field'}}}}

project_management.search_tasks: project_management.search_tasks(
task_name=None, assigned_to_email=None, list_name=None, due_date=None,
board=None) - Searches for tasks based on the given parameters.

Parameters

----------

task_name : str, optional

Name of the task.

assigned_to_email : str, optional

Email address of the person assigned to the task.

list_name : str, optional

Name of the list the task belongs to.

due_date : str, optional

Due date of the task in "YYYY-MM-DD" format.

board : str, optional

Name of the board the task belongs to.
```

Returns

-------

tasks : dict

Task information for the given parameters.

Examples

--------

>>> project_management.search_tasks("Refactor code", "tishtrya@example.com" "In progress", "2023-06-01", "Front end")

{{"task_id": "00000000", "task_name": "Refactor code", "assigned_to_email": "tishtrya@example.com", "list_name": "In Progress", "due_date": "2023-06-01", "board": "Front End"}}, args: {{'task_name': {{'title': 'Task Name'}}, 'assigned_to_email': {{'title': 'Assigned To Email'}}, 'list_name': {{'title': 'List Name'}}, 'due_date': {{'title': 'Due Date'}}, 'board': {{'title': 'Board'}}}}}

project_management.create_task: project_management.create_task(task_name=None, assigned_to_email=None, list_name=None, due_date=None, board=None) - Creates a new task.

Parameters

----------

task_name : str

Name of the task.

assigned_to_email : str

Email address of the person assigned to the task.

list_name : str

Name of the list the task belongs to.

due_date : str

Due date of the task in "YYYY-MM-DD" format.

board : str

Name of the board the task belongs to.

Returns

-------

task_id : str

8-digit ID of the new task.

Examples

--------

>>> project_management.create_task("Integrate API service with frontend",
"sam@example.com", "In progress", "2023-06-01", "Front end")

"00000001", args: {{'task_name': {{'title': 'Task Name'}},
'assigned_to_email': {{'title': 'Assigned To Email'}}, 'list_name': {{'title':
'List Name'}}, 'due_date': {{'title': 'Due Date'}}, 'board': {{'title':
'Board'}}}}

project_management.delete_task: project_management.delete_task(task_id=None) -
Deletes a task by ID.

Parameters

----------

task_id : str

8-digit ID of the task.

Returns

-------

message : str

Message indicating the status of the deletion.

Examples

--------

>>> project_management.delete_task("00000000")

"Task deleted successfully.", args: {{'task_id': {{'title': 'Task Id'}}}}

project_management.update_task: project_management.update_task(task_id=None,
field=None, new_value=None) - Updates a task by ID.

Parameters

----------

task_id : str

8-digit ID of the task.

field : str

Field to update. Available fields are: "task_name", "assigned_to_email",
"list_name", "due_date", "board"

new_value : str

New value for the field.

Returns

-------

message : str

Message indicating the status of the update.

Examples

--------

>>> project_management.update_task("00000000", "task_name", "New Task Name")

"Task updated successfully.", args: {{'task_id': {{'title': 'Task Id'}},
'field': {{'title': 'Field'}}, 'new_value': {{'title': 'New Value'}}}}

customer_relationship_manager.search_customers:
customer_relationship_manager.search_customers(customer_name=None,
customer_email=None, product_interest=None, status=None,
assigned_to_email=None, last_contact_date_min=None, last_contact_date_max=None,
follow_up_by_min=None, follow_up_by_max=None) - Searches for customers based on
the given parameters.

Parameters

----------

customer_name : str, optional

Name of the customer.

customer_email : str, optional

Email address of the customer.

product_interest : str, optional

Product interest of the customer.

status : str, optional

Current status of the customer.

assigned_to_email : str, optional

Email address of the person assigned to the customer.

last_contact_date_min : str, optional

Minimum last contact date. Format: "YYYY-MM-DD"

last_contact_date_max : str, optional

Maximum last contact date. Format: "YYYY-MM-DD"

follow_up_by_min : str, optional

Minimum follow up date. Format: "YYYY-MM-DD"

follow_up_by_max : str, optional

Maximum follow up date. Format: "YYYY-MM-DD"

Returns

-------

customers : dict

Customer information for the given parameters. Returns at most 5 records.

Examples

--------

>>> crm.search_customers(customer_name="John")

{{"customer_id": "00000001", "assigned_to_email": "sam@example.com",
"customer_name": "John Smith",

"customer_email": "john.smith@example.com", "customer_phone": "123-456-7890",
"last_contact_date": "2023-01-01",

"product_interest": "Software", "status": "Qualified", "follow_up_by": "2023-01-
15", "notes": "Had a call on 2023-01-01. "}}, args: {{'customer_name': {{
'title': 'Customer Name'}}, 'customer_email': {{'title': 'Customer Email'}},
'product_interest': {{'title': 'Product Interest'}}, 'status': {{'title':
'Status'}}, 'assigned_to_email': {{'title': 'Assigned To Email'}},
'last_contact_date_min': {{'title': 'Last Contact Date Min'}},
'last_contact_date_max': {{'title': 'Last Contact Date Max'}},
'follow_up_by_min': {{'title': 'Follow Up By Min'}}, 'follow_up_by_max': {{
'title': 'Follow Up By Max'}}}}}}

customer_relationship_manager.update_customer:
customer_relationship_manager.update_customer(customer_id=None, field=None,
new_value=None) - Updates a customer record by ID.

Parameters

----------

customer_id : str

```
ID of the customer.

field : str

Field to update. Available fields are: "customer_name", "assigned_to_email",
"customer_email", "customer_phone", "last_contact_date", "product_interest",
"status", "notes", "follow_up_by"

new_value : str

New value for the field.

Returns

-------

message : str

Message indicating the status of the update.

Examples

--------

>>> crm.update_customer("00000001", "status", "Won")

"Customer updated successfully.", args: {{'customer_id': {{'title': 'Customer
Id'}}, 'field': {{'title': 'Field'}}, 'new_value': {{'title': 'New Value'}}}}

customer_relationship_manager.add_customer:
customer_relationship_manager.add_customer(customer_name=None,
assigned_to_email=None, status=None, customer_email=None, customer_phone=None,
last_contact_date=None, product_interest=None, notes='', follow_up_by=None) -
Adds a new customer record.

Parameters

----------

customer_name : str

Name of the customer.

assigned_to_email : str

Email address of the person assigned to the customer.

status : str

Current status of the customer. One of: "Qualified", "Won", "Lost", "Lead",
"Proposal"

customer_email : str, optional

Email address of the customer.
```

customer_phone : str, optional

Phone number of the customer.

last_contact_date : str, optional

The last date the customer was contacted. Format: "YYYY-MM-DD"

product_interest : str, optional

Product interest of the customer. One of: "Software", "Hardware", "Services", "Consulting", "Training"

notes : str, optional, optional

Notes about the customer.

follow_up_by : str, optional

Date for the next follow up. Format: "YYYY-MM-DD"

Returns

-------

customer_id : str

ID of the new customer.

Examples

--------

>>> crm.add_customer("Sam Smith", "sam@example.com", "Lead", "sam.smith@example.com", "123-456-7890", "2023-01-01", "Software")

"00000201", args: {{'customer_name': {{'title': 'Customer Name'}}, 'assigned_to_email': {{'title': 'Assigned To Email'}}, 'status': {{'title': 'Status'}}, 'customer_email': {{'title': 'Customer Email'}}, 'customer_phone': {{'title': 'Customer Phone'}}, 'last_contact_date': {{'title': 'Last Contact Date'}}, 'product_interest': {{'title': 'Product Interest'}}, 'notes': {{'title': 'Notes', 'default': ''}}, 'follow_up_by': {{'title': 'Follow Up By'}}}}

customer_relationship_manager.delete_customer:
customer_relationship_manager.delete_customer(customer_id=None) - Deletes a customer record by ID.

Parameters

----------

customer_id : str

ID of the customer.

Returns

-------

message : str

Message indicating the status of the deletion.

Examples

--------

>>> crm.delete_customer("00000001")

"Customer deleted successfully.", args: {{'customer_id': {{'title': 'Customer
Id'}}}}

company_directory.find_email_address: company_directory.find_email_address(
name='') - Finds the email address of an employee by their name.

Parameters

----------

name : str, optional

Name of the person.

Returns

-------

email_address : str

Email addresses of the person.

Examples

--------

>>> directory.find_email_address_by_name("John")

"john.smith@example.com", args: {{'name': {{'title': 'Name', 'default': ''}}}}

Use a json blob to specify a tool by providing an action key (tool name) and an
action_input key (tool input).

Valid "action" values: "Final Answer" or email.get_email_information_by_id,
email.search_emails, email.send_email, email.delete_email, email.forward_email,
email.reply_email, calendar.get_event_information_by_id,
calendar.search_events, calendar.create_event, calendar.delete_event,
calendar.update_event, analytics.engaged_users_count,

```
analytics.get_visitor_information_by_id, analytics.traffic_source_count,
analytics.total_visits_count, analytics.create_plot,
analytics.get_average_session_duration,
project_management.get_task_information_by_id, project_management.search_tasks,
project_management.create_task, project_management.delete_task,
project_management.update_task, customer_relationship_manager.search_customers,
customer_relationship_manager.update_customer,
customer_relationship_manager.add_customer,
customer_relationship_manager.delete_customer,
company_directory.find_email_address
```

Provide only ONE action per $JSON_BLOB, as shown:

```
{{

"action": $TOOL_NAME,

"action_input": $INPUT

}}
```

Follow this format:

Question: input question to answer

Thought: consider previous and subsequent steps

Action:
```
$JSON_BLOB
```

Observation: action result

... (repeat Thought/Action/Observation N times)

Thought: I know what to respond

Action:
```
{{

"action": "Final Answer",

"action_input": "Final response to human"
```

```
}}
```

Begin! Reminder to ALWAYS respond with a valid json blob of a single action.
Use tools if necessary. Respond directly if appropriate. Format is Action:```$
JSON_BLOB```then Observation:.

Thought:

