# OpenReview forum: "WorkBench: a Benchmark Dataset for Agents in a Realistic Workplace Setting"
_colmweb.org/COLM/2024/Conference — COLM_

### Official Review · Reviewer_Defo · 2024-05-09

**Rating:** 6
**Confidence:** 3
**Ethics Flag:** 1

**Summary:**

The authors propose a benchmark dataset called WorkBench that consists of a variety of tasks that are typically performed in workplace settings.  Each task is outcome-centric in that the outcome is unique and unambigious to make evaluation easy and consistent. This benchmark is meant to serve as a testbed for agents to see how well they could automate business activities. They have tested 5 existing ReAct agents where Llama-2 performed poorly and GPT-4 performed significantly better and they have illustrated situations where agents can take the wrong action which could lead to disastrous outcomes (for example sending an email to the wrong person).

**Questions To Authors:**

Please see the weaknesses above

**Reasons To Accept:**

A benchmark focused on business tasks is useful for a wide variety of industries and academics
The paper is well written, easy to read, and the figures illustrate the problem well
The paper provides a nice analysis on how different agents perform based on the model across different tasks

**Reasons To Reject:**

The novelty of the paper is very weak as there are plenty of benchmarks out there that evaluate agents in similar sandbox environments including WorkArena (which tests agents in business settings as well) [1], and other outcome-centric benchmarks such as WebShop [6], MiniWob [4], WebArena [3], Mind2Web[4] , and WebVoyager [2].
This work does not include nor cite any of these benchmarks which are highly relevant.
What does this benchmark give that others lack?

If we have an agent that performs well on these aforementioned benchmarks, do we really need the proposed benchmark?

[1] WorkArena: How Capable Are Web Agents at Solving Common Knowledge Work Tasks?
[2] WebVoyager: Buildingan end-to-end web agent with large multimodal models
[3] Webarena: A realistic web environment for building
autonomous agents. A
[4] MiniWob: Reinforcement learning on web interfaces using workflowguided exploration
[5] Mind2Web: Mind2Web: Towards a generalistagent for the web.
[6] WebShop:Towards scalable real-world web interaction with grounded language agents.

---

> ### Author Rebuttal · Authors · 2024-05-30
>
> We thank the reviewer for suggesting many relevant benchmarks, which involve translating natural language tasks into actions. We will add citations in the revised version and will emphasize how WorkBench builds upon them.
>
> **WorkArena** - they have a similar evaluation approach for action tasks. They also propose a robust method for evaluating retrieval tasks, which could help build on our own work. We would like to note the following points about WorkArena:
>
> 1) The preprint first went online 12 March 2024, concurrently with our work.
> 2) WorkArena tasks focus on using a web browser, so they are often very different to tasks in our paper. For example: their task ‘Navigate to the Open" module of the "Problem" application’ is not a natural-sounding business task for a human.
> 3) WorkArena’s evaluation does not capture side effects. They do not link their domains, so tasks are limited to just one domain. The evaluation only accounts for the state of their target, rather than other parts of their system. Our evaluation considers side effects across the whole system.
>
> We will discuss WorkArena in our revision, highlighting the similarities and differences with WorkBench.
>
> **Mind2web/WebArena/WebShop/MiniWob** - these are relevant papers, with an approach that has similar benefits to our outcome-centric evaluation. Unlike WorkBench, these are focused on web browsing rather than business tasks.
>
> Another major difference is that these papers do not capture side effects. They solely evaluate the state of the target, so if the agent may take additional actions that are not desired and this would be missed in the evaluation. By comparison, WorkBench evaluates the state of all elements in our sandbox environment, not just the target element, so we can measure side effects too.
>
> We agree our evaluation is a smaller step forward compared to the web browsing literature than the papers cited in our original submission. In the revision, we will cite these web browsing papers and highlight WorkBench’s differences, particularly i) multi-step tasks that require planning ii) focus on business tasks and iii) capturing side effects.
>
> **WebVoyager** - this is a web browsing paper but without reproducible, automatic evaluation. They use human evaluators, and LLM evaluators, as “questions in our benchmark have open-ended answers”. It is still a relevant paper and we will cite it in our revised version.

---

### Official Review · Reviewer_No8N · 2024-05-10

**Rating:** 7
**Confidence:** 4
**Ethics Flag:** 1

**Summary:**

This paper proposes WorkBench, a benchmark for evaluating the ability of LLMs to condition on natural language and generate programmatic API calls ("use tools") to query and modify workplace-related databases such as calendars, emails, and project management entries. A key aspect of the benchmark is that it evaluates by matching the state of the databases, accounting for the fact that there may be multiple sequences of API calls that can correctly carry out a given task. The paper benchmarks 5 LLMs, both open- and closed-source, using the ReAct prompting strategy, and performs an error categorization.

**Questions To Authors:**

Q1) From Figure 3, it seems that the ground-truth outcome of a task is a subset of the items in the database (e.g., all events in the calendar after deleting the named event in the Fig 3 example). Is this correct that the ground-truth is a subset, or is it the entire database? If it's a subset, is the evaluation robust to the agent taking incorrect actions in other parts of the database (e.g. deleting a customer from the CRM database)? How does this interact with the "side effects" metric?

Q2) Can you say more about the "Failed to follow ReAct" subcategory of Fig 6b? This seems to be a very large fraction of the overall errors of GPT-4 on the benchmark (~16%, from combining Fig6a and Fig6b) and if this were easily fixable then performance of GPT-4 on the benchmark would be quite a bit higher.

Q3) The error analysis in Figure 6 indicates 32% "Incorrect without side effects". From the definition of side effects, does this mean that 32% had no effect on the environment (e.g., the API calls failed)?

**Reasons To Accept:**

S1) The benchmark is very well motivated: a benchmark for workplace-oriented tasks is a good step toward developing LLM agents for real-world economically impactful tasks, whereas previous works have mainly focused on consumer tasks or very specialized domains.

S2) The outcome-centric evaluation used here is an important feature of the benchmark, given that there are multiple possible ways that a task could be carried out. This addresses a key limitation of many existing benchmarks that evaluate using API call match.

S3) The paper is overall well-written (although I have a few questions about the evaluation, see below).

S4) The paper evaluated a range of reasonable models and did some interesting initial error analysis of model performance.

**Reasons To Reject:**

W1) It's a bit hard for me to gauge the difficulty of the dataset:
- The paper doesn't perform a human evaluation to see how easy or hard the examples are for people to carry out. This would be very valuable to make sure that there is some headroom beyond the performance currently achieved by GPT-4+ReAct (and see below for a concern about this setup), although given the task descriptions I expect that there will be some headroom.
- From Figure 4, ~72% of the examples require 0 or 1 actions to solve, and the performance of models on these examples is much higher (from Table 7 in the appendix) than the tasks that require more actions.
- From Figure 6, I have some doubts about how many of the errors of GPT-4 are serious errors that will be productive for future research to resolve. Many of the errors (32%) are "Incorrect without side effects", which if I understand correctly are all failures of the API calls or of the ReAct prompting itself (by simply not generating an ACTION keyword, if I understand Sec 4.4 correctly). It would be helpful to try simple baselines on top of GPT-4 that do things like re-sampling from the model if ReAct or the API calls fail. How much closer would this get to 75% (43+32) accuracy with GPT-4?

However, this isn't an absolutely crucial weakness as the benchmark is still challenging for open-source models.

W2) There are some missing comparisons to other works on LLM agent benchmarks / API calling. The below I think are the most relevant given this paper's focus on tool use and task automation in the introduction:

- AgentBench, Liu et al. 2023 (which is cited in this paper, but  in a different context)
- Mind2Web, Deng et al. 2023
- WebArena, Zhou et al. 2023
- VisualWebArena, Koh et al. 2024 (but was released in Jan 2024, close to the CoLM deadline in March)
- SMCalFlow, Andreas et al. 2020

In particular, WebArena and VisualWebArena use test-based evaluation which, like the outcome-centric evaluation proposed in this work, recognizes that multiple action paths can produce a correct outcome. So I don't think that the outcome-centric evaluation (while important!) is a key novelty of this work, if I'm understanding it correctly. But, I don't think this is a crucial weakness, as the focus on workplace-relevant tasks is important.

---

> ### Author Rebuttal · Authors · 2024-05-30
>
> **Human evaluation baseline.** This is hard to implement because humans would use GUIs, rather than APIs, to complete the tasks in WorkBench. To establish a human baseline, we could build a GUI for each tool. However, we would be testing the quality of the GUI too. Ultimately, we felt it was out of scope, but acknowledge this limit and will write about human baselines in the revision.
>
> **# of Multi-step tasks.** More may improve the paper, but the agents’ performance was already low on the current dataset. With our approach, future studies can easily create more multi-step tasks as agents improve.
>
> **Errors without side effects.** 49% are caused by not following the correct ReAct syntax. These are easier to fix than other errors via re-sampling. Fixing all of these would boost GPT-4 up to an upper bound of 59% (43+32 *.49). Re-sampling is a great suggestion and we will implement this in the revision.
>
> Other errors cannot be fixed with resampling. They are logic failures that lead to the agent not taking actions when it should. We will add this example to the revised version to clarify this error type:
>
> **Logic Failure Example** - *(Today's date, Monday 20th November)*
>
> Task: Cancel all my meetings on Tuesday
>
> Agent: Searches for meetings on 28th November.
>
> No meetings are found, so no meetings are canceled.
>
> **Web browsing papers.** We agree these are relevant, particularly WebArena, as their evaluation has similarities to WorkBench. However, a major difference is they don't capture side effects. They solely evaluate the state of the target, but the agent may take undesired actions that affect other elements. WorkBench evaluates all elements, not just the target.
>
> We agree our evaluation is a smaller step forward compared to WebArena than compared to other papers we cited. In the revision, we will discuss web browsing papers and highlight WorkBench’s differences, particularly i) multi-step tasks that require planning ii) application domains and iii) side effects.
>
> **Q1)** The ground truth is the entire database. This means we measure side effects occurring anywhere in the sandbox environment, unlike WebArena. We will clarify this in the revised paper.
>
> **Q2)** As discussed above, performance could improve by up to 16% via resampling.
>
> **Q3)** Correct, the agent either: i) Failed to make API calls e.g. because of correct syntax or invalid arguments. or ii) Used the wrong arguments, so it didn’t take actions when it should have (see example above).

---

> > ### Comment · Reviewer_No8N · 2024-06-04
> > **Thanks for the response!**
> >
> > Thanks to the authors for the response, which concisely addressed all of my main concerns. In particular, it's a great point that the proposed approach tests for unintended side effects (which many other benchmarks don't). Thanks, too, for the upper bound on GPT-4 performance and the example of the logic failure example (which, while it too could potentially be fixed with resampling, I agree is a categorically different type of error than the syntax ones) -- these were helpful!
> >
> > The major remaining limitation in my mind (as the other reviewers also pointed out) is the lack of acknowledgment of related work. But, I expect that this could easily be addressed in the camera ready especially since you have a nice description of the differences from this work in the rebuttals here.
> >
> > I've raised my score to a 7.

---

### Official Review · Reviewer_EXCB · 2024-05-10

**Rating:** 5
**Confidence:** 5
**Ethics Flag:** 1

**Summary:**

This paper proposes an evaluation framework for LLMs on goal completion, especially focusing on tool usage. 5 domains are selected, email, calendar, web analytics, crm, and projects with several functions as tools for each. They evaluated GPT, Claude, and other open models using this framework.

**Reasons To Accept:**

The paper is complete in that it provides a whole benchmark with data, tools, and metrics. It can be a valuable resource for researchers working on goal oriented Conversational AI

**Reasons To Reject:**

Interestingly the paper is totally unaware of the existing literature on this, such as the MultiWoz, SLURP, or TaskMaster like datasets, DialoGLUE, Gorilla and NexusFlow leaderboards and for the cited benchmarks they have weak arguments why the community needs Workbench. Furthermore, the framework is noly for single shot function calls, and does not support multi-turn task completion.

---

> ### Author Rebuttal · Authors · 2024-05-30
>
> Firstly, we thank the reviewer for highlighting these datasets. The main differences to emphasize are 1) our focus on practical business tasks and 2) our tasks have dependencies (e.g. needing to check the calendar before sending an email) which requires planning and actions across multiple domains. None of these papers are in a business context and none of them have tasks with dependencies.
>
> **MultiWoz** - This dataset involves updating belief states, which could be turned into function calls. However, WorkBench is different because i) We evaluate side effects, which are critical to real-world applications ii) The domains in WorkBench are highly relevant to business environments and iii) Tasks in WorkBench have dependencies that make them challenging. The agent must manage these dependencies.
>
> **SLURP** and **Taskmaster** - these focus on smart-home applications and domestic tasks respectively. While there is some overlap (e.g. "Make a calendar entry for brunch on Saturday morning"), many of their tasks are unrealistic for business applications (e.g. "brighten the lights"). Furthermore, all tasks in SLURP and TaskMaster are single-step, so none require planning.
>
> **DialoGLUE** - This is a collection of 7 other datasets (MultiWoz is one of them), rather than being a dataset itself. None involve practical business settings AND tasks that require planning across multiple domains.
>
> **Gorilla** - We believe this comment refers to the APIBench dataset introduced in [1], but please correct us if this comment refers to another paper. We cite APIBench in our paper, which evaluates API calling for models on popular ML model repositories rather than business contexts. They also use function-calling accuracy rather than outcome-centric evaluation. Finally, their tasks do not require planning.
>
> **NexusFlow**. We couldn’t find a citation for this work. If possible, could the reviewer please provide us with a citation?
>
> These papers are all relevant benchmarks for translating natural language tasks into actions. Some even have similar-sounding tasks to our work, such as “Make a calendar entry for brunch on Saturday morning” in SLURP. We will add citations for each of them.
>
> **On multi-turn dialogues**
>
> We acknowledge this could be a better format for complex requests. We will add discussion on how future work could build on WorkBench with this.
>
> We would like to clarify that many WorkBench tasks require multiple function calls, as shown in Figure 4.

---

> > ### Comment · Reviewer_EXCB · 2024-06-06
> >
> > Thanks for the rebuttal. I agree that having dependencies is valuable for dialogue evaluation. I raise my score to 5.

---

### Official Review · Reviewer_oM5y · 2024-05-12

**Rating:** 6
**Confidence:** 4
**Ethics Flag:** 1

**Summary:**

The paper proposes a new dataset called WorkBench for evaluating the how well agents can perform business-related tasks, such as sending e-mails, given a natural language instruction from the user. The instruction could be simple (such as "cancel my next meeting with Carlos") or quite complex ("if our website page views fell by more than 10%, then schedule a meeting; otherwise, send an email"). To evaluate whether the agent performed the correct actions, the authors provide a simulator for the actions and propose to measure whether executing the actions results in the correct changes to the environment (i.e. the correct updates to the databases). The authors evaluate the dataset on several language models using the ReAct framework.

**Reasons To Accept:**

- The paper addresses a timely and relevant topic.
- Even though the paper focuses on proposing a new dataset, it still evaluates the dataset on multiple models to provide relatively comprehensive results.
- The authors annotated all of the tasks with a detailed way to measure whether the task was performed correctly, ensuring a more precise evaluation.

**Reasons To Reject:**

- The outcome-centric evaluation has several limitations which seem under-studied in this work:
  - The agents may perform actions which coincidentally happen to result in the same outcomes as what the user intended given the provided databases, but would not produce the desired result when executed in a different environment.
  - QA-style tasks, not just performing actions, are still important for an agent assisting users in a workplace setting. The outcome-centric evaluation does not seem to be able to cover this use case, because it focuses on evaluating changes to the databases, and not what information is returned to the user.
- The design of the dataset focuses on complex user requests specified in one turn, which is unnatural; a dialogue-style interaction where the user can use multiple turns, and also correct errors from previous turns, would be better.
- The use of templates to create the dataset may have resulted in insufficient linguistic variation, which is not discussed in the paper.

---

> ### Author Rebuttal · Authors · 2024-05-30
>
> **The limits of outcome-centric evaluation.** Thank you for the helpful comments about this. Taking them in turn:
>
>   1. We acknowledge that our approach sometimes allows agents to get to the right answer with the wrong reasoning. However, this trades off against a major benefit: our approach enables the agent to correct its own mistakes and choose alternate, valid reasoning paths. Section 3.4 provides an example of where this is beneficial. We will discuss this trade-off explicitly to the paper and provide an example illustrating it.
>
>   2. We agree that our evaluation approach does not allow for pure QA tasks. We note that there are many other QA benchmarks, such as GAIA and ToolQA, in Section 2. A helpful extension of our work would be to add an evaluation method for QA tasks too. We will discuss this further in the final section.
>
> **Multi-turn implementation** would be an interesting extension. We will add a section to the discussion about building on our paper with multi-turn dialogues, while noting that our current implementation still allows for tasks that require planning and multiple actions.
>
> **Linguistic variation in templates** - we agree this is important. While we included 2-3 linguistic variations per template, this was not clearly communicated in the original paper. We included data in Appendix A.3 to justify the template approach, however we did not write about the linguistic variation that we included. In the revised version, we will write about this in the main body of the text, and add examples to the appendix.

---

> > ### Comment · Reviewer_oM5y · 2024-06-06
> >
> > Thank you for your thoughtful responses to the comments.
> >
> > I agree that it is highly beneficial to allow the agent to correct its own mistakes as shown in Section 3.4, but there are other ways to avoid penalizing the agent incorrectly (for example, by disregarding calls which have no effect). No single metric will be superior to others in every way so I think it can still be useful to consider having multiple metrics, or discuss the tradeoff in greater detail as you mentioned you would do.
> >
> > I agree with your other proposed additions as well.

---

> ### Author Response · Authors · 2024-06-06
> **On Linguistic Variation**
>
> Thanks for acknowledging the proposed additions will improve the paper.
>
> **On linguistic variation** - we want to provide a specific example of how this is included in the current approach. The below shows two queries from the current dataset, generated from the same template but with linguistic variation:
>
> **Example 1** "carlos is leaving the company. Can you cancel all future meetings with them?"
>
> **Example 2** "Cancel all future meetings with anaya"
>
> We were unable to include this in our last response due to space constraints. However, we think it is important to highlight given it featured as a reason to reject.
>
> We appreciate you taking the time to leave detailed feedback, which will improve the paper.

---

### Decision · Program_Chairs · 2024-07-10

**Decision:**

Accept

**Comment:**

Reviewers generally agree on the value of the WorkBench challenge set, and the authors have made available additional details about the evaluation in response to reviewer No8N in particular that should be built into the revised paper. The primary weakness of the work raised by reviewers is the lack of situated discussion within the other similar works available before as well as concurrent work that should be acknowledge in a revision. There should be space to clearly situate this work and add these additional experimental details in a revised paper.